# Spike in Asthma Healthcare Presentations in Eastern England during June 2021: A Retrospective Observational Study Using Syndromic Surveillance Data

**DOI:** 10.3390/ijerph182312353

**Published:** 2021-11-24

**Authors:** Alex J. Elliot, Christopher D. Bennett, Helen E. Hughes, Roger A. Morbey, Daniel Todkill, Ross Thompson, Owen Landeg, Emer OConnell, Mark Seltzer, Will Lang, Obaghe Edeghere, Isabel Oliver

**Affiliations:** 1Real-Time Syndromic Surveillance Team, Field Service, UK Health Security Agency, Birmingham B2 4BH, UK; chris.bennett@phe.gov.uk (C.D.B.); helen.hughes@phe.gov.uk (H.E.H.); roger.morbey@phe.gov.uk (R.A.M.); dan.todkill@phe.gov.uk (D.T.); obaghe.edeghere@phe.gov.uk (O.E.); 2National Institute for Health Research Health Protection Research Unit in Emergency Preparedness and Response, King’s College London, London SE5 9RJ, UK; 3Extreme Events and Health Protection, UK Health Security Agency, London SE1 8UG, UK; ross.thompson@phe.gov.uk (R.T.); owen.landeg@phe.gov.uk (O.L.); Emer.oconnell@phe.gov.uk (E.O.); 4National Institute for Health Research Health Protection Research Unit in Environmental Change and Health, London School of Hygiene and Tropical Medicine, London WC1H 9SH, UK; 5Met Office, Fitzroy Road, Exeter EX1 3PB, UK; mark.seltzer@metoffice.gov.uk (M.S.); will.lang@metoffice.gov.uk (W.L.); 6Chief Scientific Advisor Group, UK Health Security Agency, London SE1 8UG, UK; isabel.oliver@phe.gov.uk

**Keywords:** asthma, difficulty breathing, thunderstorm asthma, pollen, syndromic surveillance

## Abstract

Thunderstorm asthma is often characterised by a sudden surge in patients presenting with exacerbated symptoms of asthma linked to thunderstorm activity. Here, we describe a large spike in asthma and difficulty breathing symptoms observed across parts of England on 17 June 2021. The number of healthcare presentations during the asthma event was compared to expected levels for the overall population and across specific regions. Across affected geographical areas, emergency department attendances for asthma increased by 560% on 17 June compared to the average number of weekday daily attendances during the previous 4 weeks. General practitioner out of hours contacts increased by 349%, National Health Service (NHS) 111 calls 193%, NHS 111 online assessments 581% and ambulance call outs 54%. Increases were particularly noted in patient age groups 5–14 and 15–44 years. In non-affected regions, increases were small (<10%) or decreased, except for NHS 111 online assessments where there was an increase of 39%. A review of the meteorological conditions showed several localised, weak, or moderate thunderstorms specifically across parts of Southeast England on the night of June 16. In this unprecedented episode of asthma, the links to meteorologically defined thunderstorm activity were not as clear as previous episodes, with less evidence of ‘severe’ thunderstorm activity in those areas affected, prompting further discussion about the causes of these events and implications for public health management of the risk.

## 1. Introduction

Epidemics of asthma temporally associated with thunderstorm activity have been previously described and referred to as ‘thunderstorm asthma’ [1,2,3]. Thunderstorm asthma episodes are often characterised by a sudden surge in patients presenting to acute and emergency healthcare services with exacerbated symptoms of asthma. During November 2016, Melbourne, Australia reported the largest recorded episode of thunderstorm asthma where emergency department (ED) attendances for respiratory problems increased by 672% and 10 deaths were associated with the episode [4,5]. This has highlighted the potential for severe and large-scale impacts on the health of the population from these meteorological events.

The underlying drivers of thunderstorm asthma are not fully understood but are thought to include a complex mix of meteorological, environmental, and physiological factors. A variable that has been strongly associated with thunderstorm asthma is grass pollen. However, fungal spores have also been implicated [6]. Due to the presence of strong up- and down-draught air flows within thunderstorms, the pollen grains are drawn up into the clouds and storm system. Excess moisture and humidity in the clouds then infiltrate into the pollen grains (and fungal spores) causing them to rupture (osmotic shock), releasing smaller inhalable allergens carrying granules and/or other particles [7]. The excess moisture falls as rain, which deposits micro-particles from the clouds at ground level. It is these micro-particles, which can be inhaled and penetrate deep into the lungs to cause respiratory issues, which may manifest as severe asthma symptoms. Thunderstorm outflow (gust fronts) may also act to “snowplough” pollen through urban areas. The role of other meteorological conditions including humidity, atmospheric pressure and wind speed have also been explored [5].

On 17 June 2021, Public Health England (PHE) observed a large spike in health care seeking behaviour by patients presenting with asthma and difficulty breathing symptoms across parts of England. These increases were detected by real-time syndromic surveillance systems coordinated by PHE during a period when thunderstorm activity had been forecast and observed by the UK Meteorological Office (Met Office). Here, we describe the epidemiology of the asthma and difficulty breathing spikes and explore available meteorological and environmental data to understand potential causes of this episode, thereby strengthening the evidence base supporting the development of early warning alerting systems to provide better messaging to enable asthmatics to take preventative measures to reduce health risks posed by future thunderstorm asthma events.

## 2. Materials and Methods

We conducted a retrospective observational analysis of routinely collected syndromic surveillance data that informed on healthcare seeking behaviour across the National Health Service (NHS) in England. The analysis was based upon the prospective observations that had been made by PHE on 17 June 2021. Retrospectively, data for 17 June were compared with expected healthcare seeking behaviour over the previous 4 weeks.

The study aimed to describe an event that had been detected using anonymised population-level surveillance data routinely collected as part of the public health remit of PHE. It was not appropriate to involve patients or the public in the design, or conduct, or reporting, or dissemination plans of our research.

PHE (please note: on 1 October 2021, PHE was replaced by the UK Health Security Agency and Office for Health Improvement and Disparities) routinely coordinates a suite of national syndromic surveillance systems including: ED attendances; general practitioner (GP) in hours and out of hours consultations; ambulance dispatch calls; and NHS telehealth calls and online assessments (NHS 111 service) [8]. Anonymised data are extracted daily from each system, with clinical codes mapped to a suite of syndromic indicators designed to inform on particular areas of public health concern e.g., respiratory and gastrointestinal conditions or environmental impacts. Data from each syndromic system are individually monitored and interrogated on a daily basis, using epidemiological and statistical methods [9]. Anomalies (deviations from historical thresholds i.e., exceedances) requiring further epidemiological investigation are fully risk assessed (including identification of multi-system exceedances), informing public health action taken where necessary [10].

Daily counts of asthma and difficulty breathing syndromic indicators (Appendix A) were plotted and visualised, stratified at national (England), regional (PHE Centre) and age-band level. Daily counts were further aggregated into those regions affected by the Met Office severe thunderstorm forecasts and those non-affected regions. Aggregated counts in each syndromic system were compared across affected and non-affected regions for ‘expected’ activity (based on the average daily weekday count of the preceding 4 weeks, 24 May to 16 June) and ‘observed’ daily counts of the asthma and difficulty breathing indicators on 17 June.

## 3. Results

On 17 June 2021, spikes in asthma and difficulty breathing related syndromic indicators were detected across several PHE syndromic surveillance systems. Numbers of asthma and difficulty breathing cases increased sharply on 17 June, in ED attendances (asthma), NHS 111 calls and online assessments (difficulty breathing), out of hours GP contacts (difficulty breathing/wheeze/asthma) and ambulance calls (breathing problems) (Figure 1). Visually, ED attendances demonstrated the greatest increase, peaking on 17 June and then returning to expected levels over the following days. NHS 111 calls and GP out of hours contacts also spiked on 17 June, however activity did not return to normal immediately, with elevated numbers reported over the following days.

When analysed by English region, London, East of England and Southeast regions had the biggest increases (Appendix A). In those affected regions, there was a differential impact observed across different systems (Figure 1): the greatest impact was seen in ED attendances for asthma where attendances increased by 560% on 17 June compared to the average number of weekday daily attendances during the previous 4 weeks (Table 1). GP out of hours contacts increased by 349%, NHS 111 calls 193%, NHS 111 online assessments 581% and ambulance call outs 54%. In non-affected regions, increases were small (<10%) or decreased, except for NHS 111 online assessments where there was an increase of 39%.

When analysed by age group, the 5–14 and 15–44 years age groups were predominantly impacted across all systems (Appendix A). For children aged 5–14 years, GP out of hours contacts increased by 347%, ED attendances by 292% and NHS 111 calls 270%; in the 15–44 years age group GP out of hours contacts increased by 354%, NHS 111 calls by 268% and ED attendances 248% (Appendix A). For ED asthma attendances there was additionally an increase in patients aged 45–64 years (Appendix A).

A retrospective review of the meteorological conditions on the night of June 16 between 22:00 and 06:00BST (June 17) showed several localised, weak or moderate thunderstorms specifically across parts of Southeast England (Sussex, Kent, East London, Essex and East Suffolk) moving northeast-wards (Figure 2). The last notable thunderstorm had left the east coast by 06:00BST and was followed by a general widespread area of moderate to heavy rain across much of the Southeast (with no lightning), which dissipated by 11:00BST. The prevailing weather conditions overnight and into the day of June 17 were warm and humid (high dewpoints up to 16 or 17 °C, temperatures 18–21 °C) with very light winds, and likely rather damp following the overnight rain and thunderstorms.

## 4. Conclusions

The PHE national real-time syndromic surveillance service detected large spikes in asthma and difficulty breathing-type indicators on 17 June 2021. Exceedances were particularly observed in southeastern regions of England, and in school-aged children and young adults. This was one of the largest and most significant short-term population-level health impacts detected by these syndromic systems in over 20 years of operation. On 17 June, asthma attendances in EDs participating in the PHE ED syndromic surveillance system increased by over 500%, with contacts for difficulty breathing/wheeze/asthma made to GP out of hours services increasing by over 300%. Completed online assessments made to the NHS 111 online digital health tool also increased by over 500%.

Increases were observed particularly in children aged 5–14 years and young adults 15–44. The ED and GP out of hours systems also detected increases in adults aged 45–64 years. The mean age of patients attending EDs during the 2016 Melbourne thunderstorm asthma episode was 32 years while the greatest impact on hospital admissions was seen in patients aged between 20–59 years [1]. Therefore, while our results for adults seem to support findings from Melbourne, our findings illustrating the impact on children requires further investigation. Age specific factors such as sensitivity to pollen and propensity to consult a healthcare professional may influence the findings for younger cohorts of the population.

One of the key strengths of PHE’s syndromic approach is the use of multiple national systems, which in this case allowed the triangulation of results to strengthen confidence in the observed findings. Of interest, consultations to the PHE GP ‘in hours’ surveillance system did not show similar impact [11]. In the UK, for patients to access ‘routine’ GP services they are required to book an appointment. These services are also not typically available at weekends thereby resulting in increased weekend activity of other services such as NHS 111 and GP out of hours (Figure 1). Therefore, the sudden acute nature of this event suggests that patients were more likely to access unscheduled healthcare services in order that they could be seen immediately. It also points to the severe nature of symptoms, suggesting that patients affected needed to seek health advice urgently.

A limitation of syndromic surveillance is the ‘non-specific’ nature of monitoring syndromic indicators based on chief complaints, symptoms, or provisional diagnoses data. It is often difficult to attribute causes to unexpected (i.e., non-seasonal) increases in indicators. However, it is often the experience and expertise of those teams coordinating the systems to utilise other sources of intelligence to generate hypotheses to explain underlying aetiologies driving the observed increases. In this event, during the week preceding the spike in syndromic indicators the UK Met Office had released yellow national weather warnings for thunderstorm activity in parts of England, focusing mainly on flooding impacts. Viewed alongside the standard risk assessment of the syndromic data, these weather warnings prompted an initial hypothesis that the exceedances in asthma and difficulty breathing indicators were likely to be linked to thunderstorm activity.

A retrospective analysis of available meteorological conditions showed possible clues that weather conditions may have contributed to the event in some way, however it is not overly clear how, nor can they be attributed to immediate thunderstorm activity since they occurred overnight and not during the day. Weak outflow (gust fronts) was possible during the overnight thunderstorms, which may have concentrated any aeroallergens/aerosols into local pockets and close to the surface by the morning (the latter due to a stable boundary layer in the first few hundred metres of atmosphere above the surface). Other potential meteorological and environmental factors such as temperature inversions and levels of air pollution might also have contributed to the event, however relevant data were not available for inclusion in this report (but warrant further investigation).

At the time of this episode, grass pollen forecasts were very high, confirmed by national pollen forecasts and related syndromic surveillance indicators monitoring the health impact of pollen (e.g., GP consultations for allergic rhinitis and NHS 111 calls for eye problems), which were at seasonal peaks of activity [11]. Therefore, our provisional conclusions about the factors underlying the 17 June asthma episode are that thunderstorms and/or lightning activity contributed to (but were not specifically the primary variables) driving the increase in asthma and difficulty breathing presentations. It is likely that high grass pollen levels with a combination of other meteorological conditions was responsible. All other recorded episodes of thunderstorm asthma in the UK have occurred during June/July, which can be defined as the grass pollen season, and therefore this latest episode further supports the importance of grass pollen in driving these spikes in asthma [2,3,12,13,14,15,16,17,18]. This episode therefore should stimulate further discussion and research about the terminology of ‘thunderstorm asthma’ and whether these episodes should be labelled more generally as ‘storm asthma’ or ‘severe weather asthma’. Furthermore, it might be necessary to refocus research of thunderstorm asthma episodes in England, widening the scope of research to outside that of weather events consisting of thunder and lightning storms only, and to investigate the association of past thunderstorm asthma events to other meteorological factors that are linked to the thunderstorms in question, e.g., humidity, convergent cross winds/gust fronts.

Potential confounders for this incident were also considered. At the time of the episode, COVID-19 activity in England was increasing [19], however syndromic systems utilise specific clinical COVID-19 indicators rather than symptoms of asthma or difficulty breathing. While we cannot rule out that some of the presenting cases were possibly COVID-19 and ‘miscoded’ as asthma, the short-lived and sudden nature of the asthma spike does not fit clinically with the symptom progression of COVID-19, or the overall progression of the epidemic on a population level. Likewise, the incident occurred during a period of unseasonal activity of other respiratory pathogens including rhinovirus and respiratory syncytial virus [19]. However, these respiratory pathogens are more likely to be coded as ‘acute respiratory infection’ and are particularly observed in young children aged less than 5 years. Furthermore, the specific identification of short-lived spikes in activity, in a clustered but large geographical area again goes against the potential for COVID-19 or other pathogens to be responsible.

Thunderstorm asthma episodes are short lived-in nature, resulting in very sudden and short-lived increases in health care seeking behaviour. Syndromic surveillance systems are traditionally operated on a daily basis. The nature of thunderstorm asthma therefore means that any early warning or forecasting of potential events must be based upon meteorological and environmental variables. Real-time syndromic surveillance systems can inform on the subsequent health impact of thunderstorm asthma, however, there is little scope for them informing real-time forecasts. However, the value of resulting syndromic surveillance data is the retrospective analysis of previous meteorological events and thereby retrospectively identifying correlation between ‘candidate’ meteorological events and resulting health data from syndromic surveillance. This retrospective research will be used to explore whether meteorological and environmental indicators can be developed to support an early warning system for thunderstorm asthma. Currently, the UK Met Office provides severe weather warnings (yellow, amber, and red), including thunderstorms [20]. However, if these alerts could be further refined to specifically provide health advice to asthma sufferers, it could reduce the impact on the at-risk population and allow health care service providers to prepare for surges during future episodes. Such advice might include ensuring that patients carry reliever inhalers at all times; hay fever medications are taken during periods of risk; and steps are taken to avoid outdoor activity before and during storms [21].

In the UK, climate change will mean that changes in temperature and rainfall may lengthen the pollen season and potentially make pollen concentrations higher. It is also possible that climate change will lead to changes in the potency of pollen (a single pollen particle can have varying amounts of the allergy-causing agent on it). This, coupled with predictions of more severe weather events could lead to an increase in the frequency of thunderstorm asthma events in the UK. Therefore, any interventions that can be developed, tested, and implemented now will become increasingly important in protecting the population in future years. Syndromic surveillance will play an increasingly important role in helping better understand the impact of climate on health, and the importance of this in the context of climate change with the prospect of the future occurrence of more severe weather events.

## Figures and Tables

**Figure 1 ijerph-18-12353-f001:**
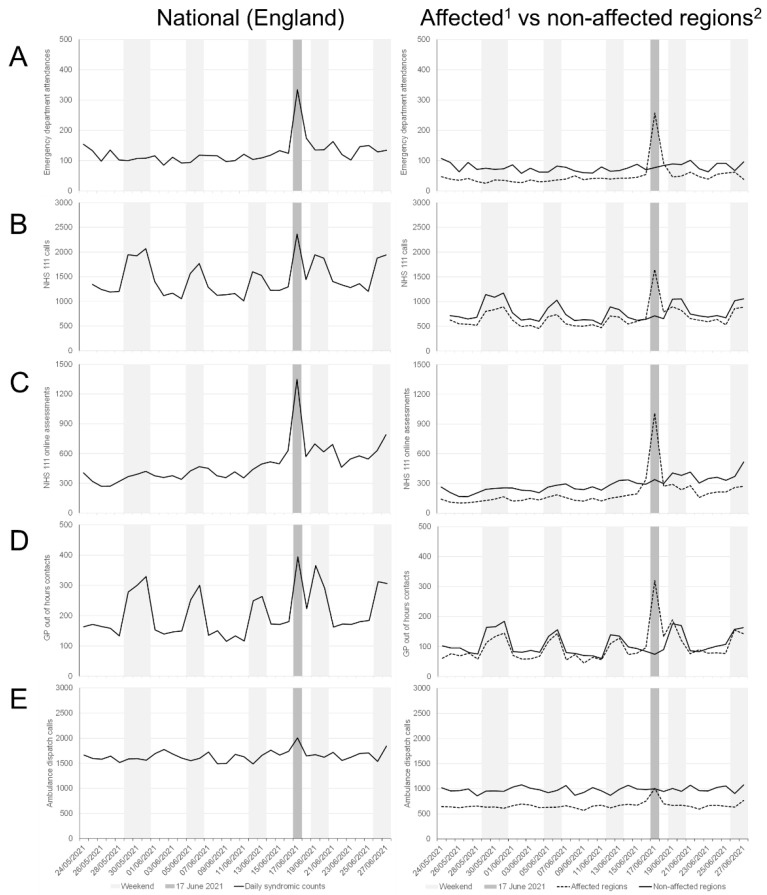
Daily (**A**) emergency department attendances for asthma; (**B**) NHS 111 calls for difficulty breathing; (**C**) NHS 111 online assessments for difficulty breathing; (**D**) out of hours general practitioner contacts for difficulty breathing/wheeze/asthma; and (**E**) ambulance calls for breathing problems for England (left panels) and stratified by storm affected and non-affected regions (right panels). ^1^ Counts aggregated across London, Southeast, East of England regions; ^2^ counts aggregated across Northwest, Northeast, Yorkshire and Humber, West Midlands, East Midlands, and Southwest regions.

**Figure 2 ijerph-18-12353-f002:**
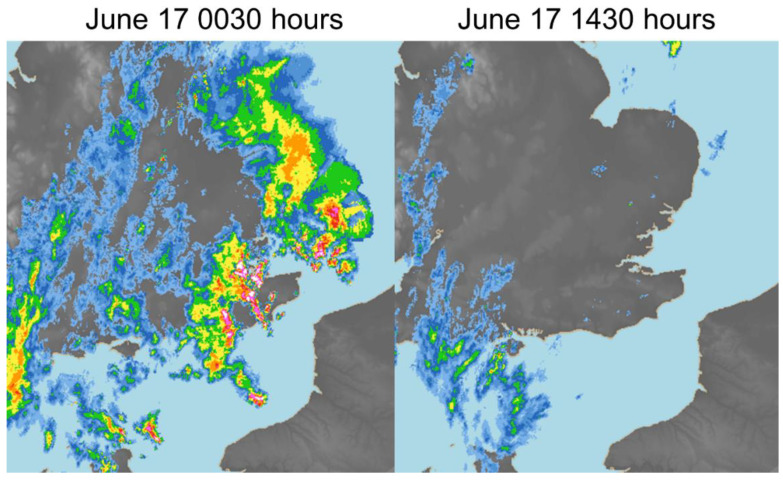
Rainfall radar for June 17 2021 0030BST (**left** panel) and 1430BST (**right** panel) centred on southeast England. Thunderstorm locations are illustrated by red and white pixels where rainfall is heaviest. Source: Met Office ^©^Crown Copyright 2021.

**Table 1 ijerph-18-12353-t001:** Observed and expected cases of asthma and difficulty breathing in across affected and non-affected regions 17 June 2021.

Syndromic System	Indicator	Affected Regions ^1^			Non-Affected Regions ^2^		
		Expected ^3^	Observed ^4^	% Increase	Expected	Observed	% Increase
Emergency department	Asthma attendances	39	257	560	76	77	2
NHS 111	Difficulty breathing calls	564	1651	193	686	712	4
NHS 111 online	Difficulty breathing assessments	148	1009	581	244	339	39
GP out of hours	Difficulty breathing/wheeze/asthma contacts	71	320	349	89	74	−16
Ambulance	Breathing problems calls	653	1006	54	984	998	1

^1^ Counts aggregated across London, Southeast, East of England regions; ^2^ counts aggregated across Northwest, Northeast, Yorkshire and Humber, West Midlands, East Midlands, and Southwest regions; ^3^ average number of daily counts expected on a weekday (Monday to Friday) based upon activity recorded in the preceding four weeks (weekdays inclusive through 24 May to 16 June 2021); ^4^ average number of daily counts observed based upon activity on 17 June 2021.

## Data Availability

The datasets used in this study are not publicly available. The aggregated data presented in Figure 1 can be made available on request through the UKHSA Office for Data Release: https://www.gov.uk/government/publications/accessing-public-health-england-data/about-the-phe-odr-and-accessing-data, accessed on 22 November 2021.

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
