# Peer review of "Spike in Asthma Healthcare Presentations in Eastern England during June 2021: A Retrospective Observational Study Using Syndromic Surveillance Data"

_ijerph, 2021, doi:10.3390/ijerph182312353_

Round 1

Reviewer 1 Report

The article contains a well written description of an epidemiologically interesting episode of asthma exacerbations on populational level in the context of a special weather condition. While I do not have objections to this article and I agree that it can be published as it is, I would nonetheless be curious to see some comments in the article regarding the B) and D) parts of figure 1. The spike that the authors report about is one of many others. It would be interesting for the readers to know what caused the other spikes in NHS 111 calls for difficulty breathing or out of hours general practitioner contacts for difficulty breathing/wheeze/asthma, encountered before 17th of June. How come that they do not translate in spikes on A, C, E? I would say that, at least for me, such a discussion would make the article even more interesting.

Kind regards,

Author Response

Reviewer 1

The article contains a well written description of an epidemiologically interesting episode of asthma exacerbations on populational level in the context of a special weather condition. While I do not have objections to this article and I agree that it can be published as it is, I would nonetheless be curious to see some comments in the article regarding the B) and D) parts of figure 1. The spike that the authors report about is one of many others. It would be interesting for the readers to know what caused the other spikes in NHS 111 calls for difficulty breathing or out of hours general practitioner contacts for difficulty breathing/wheeze/asthma, encountered before 17th of June. How come that they do not translate in spikes on A, C, E? I would say that, at least for me, such a discussion would make the article even more interesting.

Response: we thank the reviewer for this observation. The ‘spikes’ that the reviewer is referring to are weekends where traditionally activity in the NHS 111 call and GP out of hours system increases due to the closure of other NHS health services. We have made this clearer in the figures by adding vertical bars to represent weekends (line 166-168).

Reviewer 2 Report

  1. I wonder if the authors can also obtain pollen level data from ambient monitoring stations and present a comparison of pollen levels between the (thunderstorm) affected and unaffected regions. This can help to support the authors' hypothesis that grass pollen during the thunderstorm contributed to asthma exacerbations.
  2. If historical data were available, it would be very interesting to see similar spikes of asthma ED visits during previous thunderstorms over the past years, and compare them to the current one (June 17, 2021)
  3. In the Supplement, the authors present plots to show the spike of asthma ED visits by age groups, and in the main text, the authors mentioned that "when visually analyzed by age groups ...". Since data by age group are available, the by-age investigation doesn't have to stay "visually" only, and I suggest the authors present numerical results by age as well, which would make the findings and conclusions related to age more informative. This can be done by adding a supplementary table by age group that mimics Table 2, and mentioning the % increase by age group in the main text, especially for the 5-14 and 15-44 years old. 

Author Response

Reviewer 2

  1. I wonder if the authors can also obtain pollen level data from ambient monitoring stations and present a comparison of pollen levels between the (thunderstorm) affected and unaffected regions. This can help to support the authors' hypothesis that grass pollen during the thunderstorm contributed to asthma exacerbations.
  2. If historical data were available, it would be very interesting to see similar spikes of asthma ED visits during previous thunderstorms over the past years, and compare them to the current one (June 17, 2021)
  3. In the Supplement, the authors present plots to show the spike of asthma ED visits by age groups, and in the main text, the authors mentioned that "when visually analyzed by age groups ...". Since data by age group are available, the by-age investigation doesn't have to stay "visually" only, and I suggest the authors present numerical results by age as well, which would make the findings and conclusions related to age more informative. This can be done by adding a supplementary table by age group that mimics Table 2, and mentioning the % increase by age group in the main text, especially for the 5-14 and 15-44 years old. 

Response: we thank the reviewer for this review.

Response point 1: pollen data was not readily available for this report and our main aim was to disseminate the findings in a timely manner. We are however planning a more comprehensive research project investigating TSA which would involve including pollen counts as a variable.

Response point 2: using our syndromic surveillance systems we have previously published a paper describing a TSA event in London in 2015, which we have referenced in the paper. As mentioned above, we do plan a more comprehensive research project investigating TSA which would involve a retrospective look back over the years to identify any other TSA events.

Response point 3: as suggested we have now included an additional Supplementary Table S2 which presents data by age group. We have also added an additional line to the results.

Reviewer 3 Report

The end of the introduction should describe the rationale for conducting the study.  

Were increases in outcomes detected in the days following the event which was on 17 June 2021.  It is possible that the increases occurred for several days after the acute event.

The discussion should address potential mechanisms by which only certain age groups were predominately impacted.

Were pollution levels and/or weather inversions assessed as potential contributors?  Either way, these should be discussed as potential contributors. 

The authors discuss pollen forecasts.  Were any certified pollen counts available for this time period.  This is important because it is possible the counts were much higher than forecast.  If not, this shortcoming should be addressed in the discussion.

Were any certified mold counts available for this time period?  It is possible that elevated mold and grass pollen counts as well as the thunderstorms contributed to the outcomes.  If not, this shortcoming should be addressed in the discussion.

In the next to last paragraph of the discussion, consider describing what could be added to alerts in regards to health advice for asthmatic patients. 

Author Response

Reviewer 3

  1. The end of the introduction should describe the rationale for conducting the study.  
  2. Were increases in outcomes detected in the days following the event which was on 17 June 2021.  It is possible that the increases occurred for several days after the acute event.
  3. The discussion should address potential mechanisms by which only certain age groups were predominately impacted.
  4. Were pollution levels and/or weather inversions assessed as potential contributors?  Either way, these should be discussed as potential contributors. 
  5. The authors discuss pollen forecasts.  Were any certified pollen counts available for this time period.  This is important because it is possible the counts were much higher than forecast.  If not, this shortcoming should be addressed in the discussion.
  6. Were any certified mold counts available for this time period?  It is possible that elevated mold and grass pollen counts as well as the thunderstorms contributed to the outcomes.  If not, this shortcoming should be addressed in the discussion.
  7. In the next to last paragraph of the discussion, consider describing what could be added to alerts in regards to health advice for asthmatic patients. 

Response: we thank the reviewer for these constructive comments.

Response point 1: we have added an additional line to explain the rationale for conducting the study

Response point 2: yes, increases persisted after the event, for differing periods by each system. We have added a line in the discussion to highlight this.

Response point 3: this is an important point and we have added some additional discussion.

Response point 4: there were not included in the paper as relevant data were not available. However, we agree that these are important points and further discussion has been included to reflect this.

Response point 5: pollen forecasts are currently available in the UK however corresponding count data are not available openly. We are however planning a more comprehensive research project investigating TSA which would involve including pollen counts as a variable.

Response point 6: spore forecasts are not currently available in the UK. Spore collection data is collected however there are only a few sites across the country. In the more comprehensive research project investigating TSA we would also explore the availability of retrospective spore data to include as a variable in the model.

Response point 7: we agree that this is important – we have included some of the existing advice reference from the asthma UK charity website.

Reviewer 4 Report

Local climate change and contemporary health issues are essential areas that must be considered from a public health perspective. Elliot et al. identified and presented the asthma spike phenomenon in Eastern England dynamically from the perspective of thunderstorm asthma.

It would be better to address more epidemiologic concerns for scientific fidelity before the acceptance of the manuscript.

  1. Breathing difficulties can occur in various clinical situations. Please clearly present the definition of asthma used in this study.
  2. If the authors have any data on those with and without an existing asthma history, please provide them.
  3. Please present the abbreviation in Abstract section after the first word comes out. (ED, GP, NHS et al)
  4. If there are previous studies in other regions of the Earth experiencing similar environmental changes, it will be helpful to introduce and compare and analyze them.

Author Response

Reviewer 4

Local climate change and contemporary health issues are essential areas that must be considered from a public health perspective. Elliot et al. identified and presented the asthma spike phenomenon in Eastern England dynamically from the perspective of thunderstorm asthma.

It would be better to address more epidemiologic concerns for scientific fidelity before the acceptance of the manuscript.

  1. Breathing difficulties can occur in various clinical situations. Please clearly present the definition of asthma used in this study.

Response: each NHS healthcare service uses a different underlying clinical coding system. Therefore, the respective syndromic surveillance systems are corresponding asthma indicators are based upon different coding systems and codes. We have added an additional table in the supplementary materials (Supplementary Table S1) to present the coding systems and codes where appropriate.

  1. If the authors have any data on those with and without an existing asthma history, please provide them.

Response: unfortunately, syndromic surveillance systems do not allow for investigation of individual patient histories and therefore this is not possible.

  1. Please present the abbreviation in Abstract section after the first word comes out. (ED, GP, NHS et al)

Response: we have included full explanations of all abbreviations

  1. If there are previous studies in other regions of the Earth experiencing similar environmental changes, it will be helpful to introduce and compare and analyze them.

Response: we have included a range of references that describe previous thunderstorm asthma events. There are few specific publications on the impact of climate change and thunderstorm asthma.
